# Determinants of sustained stabilization of beta-cell function following short-term insulin therapy in type 2 diabetes

Ravi Retnakaran [1,2,3] ✉, Jiajie Pu[1], Alexandra Emery[1], Stewart B. Harris[4], Sonja M. Reichert[4], Hertzel C. Gerstein [5], Natalia McInnes[5], Caroline K. Kramer[1,2,3] & Bernard Zinman[1,2,3]

In early type 2 diabetes, the strategy of "induction" with short-term intensive insulin therapy followed by "maintenance" with metformin can stabilize pancreatic beta-cell function in some patients but not others. We thus sought to elucidate determinants of sustained stabilization of beta-cell function. In this secondary analysis of ClinicalTrials.Gov NCT02192424, adults with ≤5-years diabetes duration were randomized to 3-weeks induction insulin therapy (glargine/lispro) followed by metformin maintenance either with or without intermittent 2-week courses of insulin every 3-months for 2-years. Sustained stabilization (higher beta-cell function at 2-years than at baseline) was achieved in 55 of 99 participants. Independent predictors of sustained stabilization were the change in beta-cell function during induction and changes in hepatic insulin resistance and alanine aminotransferase during maintenance. Thus, initial reversibility of beta-cell dysfunction during induction and subsequent preservation of hepatic insulin sensitivity during maintenance are associated with sustained stabilization of beta-cell function following short-term insulin and metformin.

ClinicalTrials.Gov NCT02192424

The natural history of type 2 diabetes (T2DM) is characterized by the progressive deterioration of pancreatic beta-cell function over time[1]. Despite the genetic underpinnings of beta-cell dysfunction, there exists a reversible component to this process early in the course of T2DM that may be responsive to interventions such as intensive lifestyle modification and short-term intensive insulin therapy (IIT)[2,3]. Indeed, by ameliorating reversible beta-cell dysfunction, these interventions can even induce a remission of diabetes[2–7]. Ultimately, however, the durability of such an outcome is dependent upon the ability to maintain the beneficial beta-cell effect that was induced with the initial intervention. Thus, there is currently interest in developing treatment strategies that can achieve the sustained stabilization of beta-cell function in early T2DM and understanding the underlying determinants of this effect[8,9].

Here we show that one such strategy is the administration of initial short-term IIT (as induction therapy) followed by metformin (as maintenance therapy), an approach that has been shown to stabilize beta-cell function for 2-years in some patients but not others[10,11]. We recently reported the main findings of the REmission Studies Evaluating Type 2 Diabetes - Intermittent Insulin Therapy Main (RESET-IT Main) trial, which demonstrated that the addition of intermittent courses of IIT does not further enhance the beneficial effect on beta-cell function achieved with initial induction IIT followed by metformin maintenance[11]. Of note, this trial population underwent detailed characterization of metabolic function (including insulin sensitivity and beta-cell function) on 10 occasions over 2-years, thereby providing an opportunity to evaluate the metabolic response over time and

[1]Leadership Sinai Centre for Diabetes, Mount Sinai Hospital, Toronto, Canada. [2]Division of Endocrinology, University of Toronto, Toronto, Canada. [3]Lunenfeld-Tanenbaum Research Institute, Mount Sinai Hospital, Toronto, Canada. [4]Department of Family Medicine, Schulich School of Medicine and Dentistry, Western University, London, Canada. [5]Division of Endocrinology, McMaster University, Hamilton, Canada. ✉e-mail: ravi.retnakaran@sinaihealth.ca

**Table 1 | Baseline characteristics of study population stratified into (i) participants who had sustained stabilization of beta-cell function for 2 years and (ii) those who did not**

| | No Stabilization (n = 44) | Stabilization (n = 55) | P |
|---|---|---|---|
| Age (years) | 56.9 (16.6) | 61.2 (18.0) | 0.22 |
| Sex (% male) | 19 (43) | 33 (60) | 0.14 |
| Ethnicity: | | | 0.89 |
| White n(%) | 32 (73) | 41 (75) | |
| South Asian n(%) | 5 (11) | 8 (15) | |
| East Asian n(%) | 4 (9) | 3 (5) | |
| Other n(%) | 3 (7) | 3 (5) | |
| Duration of diabetes (years) | 1.4 (1.3) | 1.9 (1.5) | 0.08 |
| Metformin monotherapy prior to study n(%) | 28 (64) | 33 (60) | 0.87 |
| Total physical activity: | 7.5 (1.4) | 7.1 (1.2) | 0.22 |
| Sport index | 2.4 (0.7) | 2.3 (0.7) | 0.41 |
| Leisure time index | 2.6 (0.6) | 2.6 (0.6) | 0.60 |
| Work index | 2.5 (0.7) | 2.4 (0.6) | 0.56 |
| Body mass index (kg/m$^2$) | 34.2 (7.8) | 33.4 (6.7) | 0.59 |
| Waist circumference (cm) | 107.4 (12.0) | 107.6 (15.0) | 0.94 |
| ALT (IU/l) | 29 (14) | 36 (18) | 0.05 |
| Creatinine (μmol/l) | 70 (14) | 74 (16) | 0.27 |
| Glycemia: | | | |
| Fasting glucose (mmol/l) | 6.9 (1.1) | 7.9 (1.6) | **0.0003** |
| 2 h glucose (mmol/l) | 13.2 (3.7) | 14.9 (3.4) | **0.02** |
| A1c (%) | 6.4 (0.5) | 6.7 (0.7) | **0.02** |
| Insulin sensitivity/resistance: | | | |
| Matsuda index | 1.9 (1.3–2.7) | 1.6 (1.4–2.1) | 0.35 |
| HOMA-IR | 4.8 (2.9–7.3) | 5.9 (4.5–7.9) | 0.11 |
| Beta-cell function: | | | |
| ISSI-2 | 191 (145–261) | 141 (95–205) | **0.004** |
| Insulinogenic index/HOMA-IR | 1.5 (1.0–2.4) | 1.1 (0.6–1.8) | **0.04** |
| $\Delta$Cpeptide$_{0-120}$/$\Delta$gluc$_{0-120}$ × Matsuda index | 855 (417–1583) | 519 (342–969) | **0.02** |
| $\Delta$ISR$_{0-120}$/$\Delta$gluc$_{0-120}$ × Matsuda index | 3.2 (1.4–6.1) | 1.8 (1.0–3.6) | **0.03** |

Continuous variables presented as mean followed by standard deviation in parentheses (if normal distribution) or median followed by interquartile range (if skewed distribution). Categorical variables presented as proportions. Groups were compared by Analysis of Variance (normally distributed variables), Kruskal-Wallis test (skewed variables), or Chi-Square test (categorical variables). All tests were two-sided, with no adjustment for multiple comparisons. Bold indicates $p < 0.05$.

determinants of its durability. Thus, in this context, we now sought to elucidate determinants of sustained stabilization of beta-cell function in response to short-term IIT and metformin in early T2DM. We demonstrate herein that (i) initial reversibility of beta-cell dysfunction during induction and (ii) subsequent preservation of hepatic insulin sensitivity during maintenance are associated with the sustained stabilization of beta-cell function in response to short-term IIT and metformin.

## Results

The RESET-IT Main trial (ClinicalTrials.Gov NCT02192424) was a multi-centre, parallel arm trial in which adults with T2DM were randomly assigned to receive induction therapy with a 3-week course of IIT, followed by maintenance therapy consisting of either (i) metformin alone or (ii) metformin with intermittent 2-week courses of short-term IIT administered every 3-months for 2-years. As recently reported[11], the primary outcome of baseline-adjusted beta-cell function at 2-years (measured by Insulin Secretion-Sensitivity Index-2 (ISSI-2)) did not differ between the two treatment arms. The current secondary analysis focused on sustained stabilization of beta-cell function in this study population, defined by higher ISSI-2 at 2-years (or last study visit) than at baseline (i.e., reflecting an improvement in beta-cell function, in contrast to the deterioration over time that typically characterizes the natural history of T2DM).

### Characteristics at Baseline and after 3-weeks Induction

Table 1 shows the baseline characteristics of the study population stratified into the following 2 groups: (i) participants who had sustained stabilization of beta-cell function for 2-years (n = 55) and (ii) those who did not have such stabilization (n = 44). At baseline, there were no differences between these groups in age, sex, ethnicity, duration of diabetes, metformin monotherapy before the trial, BMI, waist or insulin sensitivity/resistance (Matsuda index, Homeostasis Model of Assessment of Insulin Resistance (HOMA-IR)). However, those who went on to achieve stabilization had higher A1c (mean 6.7% vs 6.4%, $p = 0.02$) than their peers, coupled with greater glycemia on the oral glucose tolerances test (OGTT) (fasting glucose: $p = 0.0003$; 2 h glucose: $p = 0.02$). Consistent with these findings, the stabilization group had lower beta-cell function at baseline than the non-stabilization group (ISSI-2: $p = 0.004$; insulinogenic index/HOMA-IR: $p = 0.04$; $\Delta$Cpeptide$_{0-120}$/$\Delta$glucose$_{0-120}$ × Matsuda: $p = 0.02$; $\Delta$ISR$_{0-120}$/$\Delta$gluc$_{0-120}$ × Matsuda: $p = 0.03$).

During the 3-weeks of induction IIT, glycemic control was similar in the 2 groups, as evidenced by the profile of mean capillary glucose measurements across the day (Supplementary Fig. 1). Nevertheless, after 3-weeks of induction IIT, there were no longer significant differences between the two groups in either glycemic measures or beta-cell function (Table 2). Indeed, compared to their peers, those who went on to achieve stabilization had greater recovery of beta-cell function (change from baseline at 3-weeks: ISSI-2 $p = 0.004$; $\Delta$Cpeptide$_{0-120}$/$\Delta$glucose$_{0-120}$ × Matsuda $p = 0.007$; $\Delta$ISR$_{0-120}$/$\Delta$gluc$_{0-120}$ × Matsuda $p = 0.02$), coupled with greater decrease in fasting glucose ($p = 0.02$) and 2-h glucose on the OGTT ($p = 0.0002$). Thus, the stabilization group had poorer beta-cell function at baseline that was responsive to induction IIT, yielding no significant differences between the 2 groups at 3-weeks (Table 2).

### Characteristics during maintenance phase

We next examined the maintenance phase (from 3-weeks to 2-years). Of note, the stabilization and non-stabilization groups did not differ in the maintenance therapy to which participants were randomized (Table 2). Similarly, time to initial loss of stabilization of beta-cell function did not differ between participants randomized to metformin alone and those randomized to metformin with intermittent IIT (log-rank $p = 0.46$; Supplementary Fig. 2). At 2-years, those who achieved stabilization had better beta-cell function than their peers (all 4 measures: $p \leq 0.02$), coupled with better whole-body insulin sensitivity (Matsuda index: $p = 0.0008$), lower hepatic insulin resistance (HOMA-IR: $p = 0.00004$) and lower 2 h glucose on the OGTT ($p = 0.03$) (Table 3). Thus, 2 groups that were comparable at 3-weeks had significant differences at 2-years that did not appear to be attributable to their maintenance therapy, thereby raising the question of potential differential changes over the intervening time.

We first examined progressive changes over time in the glucose response to the OGTT in the 2 groups (Supplementary Fig. 3). At baseline, the stabilization group had higher glucose levels at all 5 timepoints on the OGTT (Panel A) but these differences were eliminated after 3-weeks of induction IIT (Panel B). The glucose profile then did not differ between the groups at 6-months (Panel C) and 12-months

**Table 2 | Characteristics after 3-weeks of induction intensive insulin therapy (IIT) in (i) participants who had sustained stabilization of beta-cell function for 2 years and (ii) those who did not**

| | No Stabilization (n = 44) | Stabilization (n = 55) | P |
|---|---|---|---|
| **Induction IIT:** | | | |
| Initial daily basal insulin dose (units/kg) | 7.6 (2.2) | 8.7 (2.7) | **0.02** |
| Initial daily meal insulin dose (units/kg) | 10.3 (2.9) | 11.2 (4.6) | 0.23 |
| Final daily basal insulin dose (units/kg) | 20.1 (16.2) | 20.8 (14.1) | 0.82 |
| Final daily meal insulin dose (units/kg) | 16.0 (9.8) | 17.0 (12.2) | 0.64 |
| **After 3-weeks of Induction IIT:** | | | |
| Body mass index (kg/m²) | 34.0 (7.8) | 33.0 (6.7) | 0.51 |
| Waist circumference (cm) | 105.2 (11.6) | 106.4 (14.5) | 0.66 |
| ALT (IU/l) | 27 (10) | 31 (15) | 0.11 |
| Creatinine (µmol/l) | 74 (14) | 74 (14) | 0.98 |
| Glycemia: | | | |
| Fasting glucose (mmol/l) | 6.0 (1.2) | 6.3 (1.2) | 0.19 |
| 2 h glucose (mmol/l) | 13.7 (3.0) | 13.4 (3.5) | 0.65 |
| A1c (%) | 6.1 (0.4) | 6.3 (0.6) | 0.05 |
| Insulin sensitivity/resistance: | | | |
| Matsuda index | 2.4 (1.6–3.4) | 2.7 (1.7–4.0) | 0.47 |
| HOMA-IR | 3.3 (2.4–5.4) | 3.1 (2.2–5.3) | 0.68 |
| Beta-cell function: | | | |
| ISSI-2 | 206 (167–282) | 199 (151–269) | 0.52 |
| Insulinogenic index/ HOMA-IR | 1.8 (1.2–2.5) | 1.9 (1.1–2.2) | 0.59 |
| $\Delta$Cpeptide$_{0-120}$/$\Delta$gluc$_{0-120}$ × Matsuda index | 798 (515–1475) | 828 (576–1420) | 0.84 |
| $\Delta$ISR$_{0-120}$/$\Delta$gluc$_{0-120}$ × Matsuda index | 3.2 (1.9–5.2) | 2.8 (1.7–5.3) | 0.80 |
| **Change from baseline to 3-weeks:** | | | |
| $\Delta$ in body mass index (kg/m²) | –0.1 (0.6) | –0.3 (1.2) | 0.44 |
| $\Delta$ in waist circumference (cm) | –1.4 (4.1) | –1.0 (2.3) | 0.53 |
| $\Delta$ in ALT (IU/l) | –1 (8) | –4 (11) | 0.16 |
| $\Delta$ in creatinine (µmol/l) | 3 (6) | 2 (6) | 0.25 |
| Glycemia: | | | |
| $\Delta$ in fasting glucose (mmol/l) | –0.9 (1.3) | –1.6 (1.6) | **0.02** |
| $\Delta$ in 2 h glucose | 0.5 (2.5) | –1.5 (2.7) | **0.0002** |
| $\Delta$ in A1c (%) | –0.3 (0.2) | –0.3 (0.4) | 0.52 |
| Insulin sensitivity/resistance: | | | |
| $\Delta$ in Matsuda index | 0.5 (1.2) | 1.0 (1.3) | 0.06 |
| $\Delta$ in HOMA-IR | –1.6 (2.6) | –2.8 (4.5) | 0.09 |
| Beta-cell function: | | | |
| $\Delta$ in ISSI-2 | 20 (76) | 68 (84) | **0.004** |
| $\Delta$ in Insulinogenic index/ HOMA-IR | 0.3 (1.7) | 0.7 (1.2) | 0.19 |
| $\Delta$ in $\Delta$Cpeptide$_{0-120}$/ $\Delta$gluc$_{0-120}$ × Matsuda index | –559 (2437) | 1027 (3120) | **0.007** |
| $\Delta$ in $\Delta$ISR$_{0-120}$/$\Delta$gluc$_{0-120}$ × Matsuda index | –1.4 (7.7) | 2.3 (5.8) | **0.02** |
| **Maintenance Therapy Started at 3-weeks:** | | | 0.12 |
| Metformin alone (%) | 17 (39) | 31 (56) | |
| Metformin + Intermittent IIT (%) | 27 (61) | 24 (44) | |

Continuous variables presented as mean followed by standard deviation in parentheses (if normal distribution) or median followed by interquartile range (if skewed distribution). Groups were compared by Analysis of Variance (normally distributed variables), Kruskal-Wallis test (skewed variables), or Chi-Square test (categorical variables. All tests were two-sided, with no adjustment for multiple comparisons. Bold indicates $p < 0.05$.

(Panel D). However, at 18-months and 24-months, the glucose response was now higher in the non-stabilization group than in those who achieved stabilization (Panels E and F, respectively), suggestive of underlying differential changes in metabolic function.

Comparison of the absolute changes in metabolic parameters between 3-weeks and 2-years showed differences between the groups in adiposity (BMI: $p = 0.004$; waist circumference: $p = 0.03$), ALT ($p = 0.01$), glycemia (fasting glucose: $p = 0.02$; 2 h glucose: $p = 0.008$; A1c: $p = 0.0001$), HOMA-IR ($p = 0.0008$), ISSI-2 ($p = 0.0004$) and insulinogenic index/HOMA-IR ($p = 0.00009$) (Table 3). To determine whether these measures were changing differentially over time between the 2 groups, we next performed generalized least squares regression analyses (Fig. 1). As anticipated, there were differential changes over time in ISSI-2 ($p = 0.0001$) and insulinogenic index/ HOMA-IR ($p = 0.0004$) during maintenance from 3-weeks to 2-years, with the non-stabilization group showing loss of beta-cell function (Fig. 1a, b). Of note, BMI ($p = 0.033$) also changed differentially between the groups, with decrease over time evident in the stabilization group, although waist circumference ($p = 0.32$) did not show such differential change (Fig. 1c, d). However, there were differential changes in both whole-body sensitivity (Matsuda index: $p = 0.044$) and hepatic insulin resistance (HOMA-IR: $p = 0.001$), with clear deterioration of each in the non-stabilization group (Fig. 1e, f).

## Determinants of sustained beta-cell stabilization

Finally, we performed logistic regression analyses to identify independent determinants of the stabilization of beta-cell function at 2-years (Table 4). In a core model adjusted for age, duration of diabetes, baseline ISSI-2, and change in waist circumference from 3-weeks to 2-years, both change in ISSI-2 during induction and change in Matsuda index during maintenance emerged as independent predictors of stabilization of beta-cell function (Model I). Upon substitution of change in waist circumference with change in BMI from 3-weeks to 2-years, the latter measure (change in BMI) replaced the concurrent change in Matsuda index as a significant predictor (Model II). Compared to Model I, this change improved model performance (area-under-the-curve (AUC) increased and Akaike Information Criterion (AIC) decreased). These model parameters were further improved with replacement of the change in Matsuda index with the change in HOMA-IR from 3-weeks to 2-years (Model III). With this model, change in ISSI-2 during induction remained an independent predictor of stabilization (aOR = 1.02 [95%CI 1.00–1.03], $p = 0.005$) while the change in HOMA-IR during maintenance from 3-weeks to 2-years replaced the concurrent change in BMI as a significant predictor (aOR = 0.64 [0.49–0.83], $p = 0.0007$).

We next performed sensitivity analyses to evaluate the robustness of these findings from the core Model III (Table 4). These analyses revealed that the addition of total physical activity over the 2-years did not change the significant predictors (Model IV). However, upon its addition to the model (Model V), the change in ALT from 3-weeks to 2-years emerged as an additional predictor of stabilization of beta-cell function at 2-years (aOR = 0.92 [0.86–0.99], $p = 0.02$), coupled with the concomitant change in HOMA-IR during maintenance (aOR = 0.54 [0.38–0.77], $p = 0.0007$) and the change in ISSI-2 during induction (aOR = 1.01 [1.00–1.03], $p = 0.02$). Moreover, this model demonstrated better performance than the core Model III (likelihood ratio test $p = 0.015$).

On additional sensitivity analyses, we repeated the core logistic regression analyses by replacing ISSI-2 with each of the other 3 measures of beta-cell function. The significant predictors from the core model above persisted when ISSI-2 was replaced as the measure of beta-cell function with insulinogenic index/HOMA-IR, $\Delta$Cpeptide$_{0-120}$/ $\Delta$glucose$_{0-120}$ × Matsuda and $\Delta$ISR$_{0-120}$/$\Delta$gluc$_{0-120}$ × Matsuda, respectively, with the sole exception being that the change in insulinogenic

**Table 3 | Characteristics at completion of maintenance therapy at 2-years in (i) participants who had sustained stabilization of beta-cell function and (ii) those who did not**

| | No Stabilization (n = 44) | Stabilization (n = 55) | P |
|---|---|---|---|
| **At 2-years** | | | |
| Total physical activity: | 7.5 (1.5) | 7.5 (1.6) | 0.86 |
| Sport index | 2.4 (0.8) | 2.4 (0.9) | 0.98 |
| Leisure time index | 2.7 (0.6) | 2.6 (0.7) | 0.51 |
| Work index | 2.5 (0.7) | 2.5 (0.7) | 0.75 |
| Body mass index (kg/m$^2$) | 33.6 (8.3) | 32.3 (6.9) | 0.42 |
| Waist circumference (cm) | 105.9 (12.2) | 104.8 (16.2) | 0.72 |
| ALT (IU/l) | 31 (18) | 25 (14) | 0.10 |
| Creatinine (μmol/l) | 75 (15) | 75 (17) | 0.84 |
| Glycemia: | | | |
| Fasting glucose (mmol/l) | 7.1 (1.2) | 6.6 (1.4) | 0.08 |
| 2 h glucose (mmol/l) | 14.1 (4.1) | 12.2 (3.5) | **0.03** |
| A1c (%) | 6.5 (0.7) | 6.3 (0.6) | 0.13 |
| Insulin sensitivity/ resistance: | | | |
| Matsuda index | 1.5 (1.2–2.3) | 2.4 (1.8–3.1) | **0.0008** |
| HOMA-IR | 6.1 (4.1–8.9) | 3.8 (2.7–4.7) | **0.00004** |
| Beta-cell function: | | | |
| ISSI-2 | 150 (113–205) | 189 (145–312) | **0.02** |
| Insulinogenic index/ HOMA-IR | 1.0 (0.7–1.5) | 1.8 (1.1–3.4) | **0.0006** |
| $\Delta$Cpeptide$_{0-120}$/ $\Delta$gluc$_{0-120}$ × Matsuda index | 634 (330–923) | 1174 (583–1933) | **0.002** |
| $\Delta$ISR$_{0-120}$/$\Delta$gluc$_{0-120}$ × Matsuda index | 2.0 (1.1–2.8) | 3.3 (1.9–7.2) | **0.01** |
| **Change from 3-weeks to 2-years** | | | |
| $\Delta$ in body mass index (kg/m$^2$) | –0.1 (1.5) | –1.2 (1.8) | **0.004** |
| $\Delta$ in waist circumference (cm) | 0.9 (6.1) | –2.1 (5.7) | **0.03** |
| $\Delta$ in ALT (IU/l) | 2 (14) | –5 (11) | **0.01** |
| $\Delta$ in creatinine (μmol/l) | 0 (9) | –1 (8) | 0.79 |
| Glycemia: | | | |
| $\Delta$ in plasma glucose (mmol/l) | 1.1 (1.2) | 0.4 (1.3) | **0.017** |
| $\Delta$ in 2 h glucose | 0.6 (2.7) | –1.2 (3.5) | **0.008** |
| $\Delta$ in A1c (%) | 0.4 (0.5) | 0 (0.5) | **0.0001** |
| Insulin sensitivity/ resistance: | | | |
| $\Delta$ in Matsuda index | –1.1 (1.6) | –0.4 (1.7) | 0.06 |
| $\Delta$ in HOMA-IR | 3.1 (3.9) | 0.1 (3.9) | **0.0008** |
| Beta-cell function: | | | |
| $\Delta$ in ISSI-2 | –74 (73) | 9 (116) | **0.0004** |
| $\Delta$ in Insulinogenic index/ HOMA-IR | –1.1 (1.5) | 0.5 (2.0) | **0.00009** |
| $\Delta$ in $\Delta$Cpeptide$_{0-120}$/ $\Delta$gluc$_{0-120}$ × Matsuda | –497 (718) | –370 (4097) | 0.85 |
| $\Delta$ in $\Delta$ISR$_{0-120}$/$\Delta$gluc$_{0-120}$ × Matsuda index | –2.0 (2.3) | –0.5 (9.6) | 0.14 |

Continuous variables presented as mean followed by standard deviation in parentheses (if normal distribution) or median followed by interquartile range (if skewed distribution). Groups were compared by Analysis of Variance (normally distributed variables) or Kruskal-Wallis test (skewed variables). All tests were two-sided, with no adjustment for multiple comparisons. Bold indicates $p < 0.05$.

index/HOMA-IR during induction did not achieve significance. Recognizing that the current analysis was performed in the 99 study participants (out of 108) who completed at least 1-year of follow-up, we repeated the logistic regression analyses in all participants who completed at least 1 study visit after receiving any maintenance therapy (i.e., completed at least the 3-month visit). On these sensitivity analyses in 105 study participants, the significant variables from Table 4 persisted as predictors of stabilization of beta-cell function (defined by the last study visit, rather than at 2-years).

## Discussion

In this study, we show that 55 of 99 patients with T2DM achieved sustained stabilization of beta-cell function over 2-years in response to induction IIT followed by metformin maintenance. These individuals had poorer beta-cell function than their peers at baseline but experienced greater recovery thereof in response to 3-weeks of induction IIT. Over the subsequent 2-years, they had greater stability of weight/ adiposity, hepatic insulin sensitivity and ALT compared to their peers, and ultimately exhibited better beta-cell function and glycemic control. Logistic regression analyses revealed 3 independent predictors of the sustained stabilization of beta-cell function: the change in ISSI-2 during induction IIT and the changes in HOMA-IR and ALT during maintenance. It thus emerges that the initial reversibility of beta-cell dysfunction during induction and the subsequent preservation of hepatic insulin sensitivity during maintenance are underlying determinants of sustained stabilization of beta-cell function in response to short-term IIT and metformin.

Two elements of these data point to the role of reversible beta-cell dysfunction. First, it is notable that those who ultimately achieved stabilization 2-years later had poorer (not better) beta-cell function than their peers at baseline. Second, in response to induction IIT, they experienced a greater improvement in beta-cell function than did their peers. Importantly, when both of these variables were included in the optimal model (Model V in Table 4 Panel B), only the change in ISSI-2 from baseline to 3-weeks was a significant independent predictor of sustained stabilization of beta-cell function. In other words, the initial recovery of beta-cell function following induction superseded the baseline measure in predicting stabilization 2-years later. These findings are consistent with the reported role of reversible beta-cell dysfunction in determining diabetes remission with lifestyle intervention[2]. Currently, the presence of reversible dysfunction is only identifiable retrospectively since existing measures of beta-cell function cannot distinguish between reversible and irreversible components. Growing recognition of the importance of reversible beta-cell dysfunction, as supported by the findings herein, underscores the need for the identification of phenotypic markers that may indicate its presence at baseline. Indeed, such insight could prospectively inform the judicious targeting of interventions that can ameliorate reversible beta-cell dysfunction (such as intensive lifestyle modification and short-term IIT).

After addressing reversible beta-cell dysfunction with induction therapy, the next challenge is maintaining this beneficial effect. In this regard, current thinking holds that improving insulin sensitivity (e.g. through weight loss) can off-load the secretory demands placed on the beta-cells and thereby reduce their functional deterioration in T2DM. Conversely, weight gain may increase insulin resistance and hasten beta-cell demise. In this context, it is notable that, during maintenance from 3-weeks to 2-years, there were differential changes over time in BMI between the stabilization and non-stabilization groups, with evidence of a decrease in the stabilization group (Fig. 1C). However, far more striking were the differential changes in insulin sensitivity/ resistance, possibly reflecting downstream sequelae of the changes in BMI. Indeed, during this time, the non-stabilization group experienced deterioration of whole-body insulin sensitivity (Matsuda index) and hepatic insulin resistance (HOMA-IR), in contrast to the relative stability of these measures in the stabilization group (Fig. 1E, F). Furthermore, the sequential development of the core logistic regression model in Table 4 Panel A shows that the change in HOMA-IR from

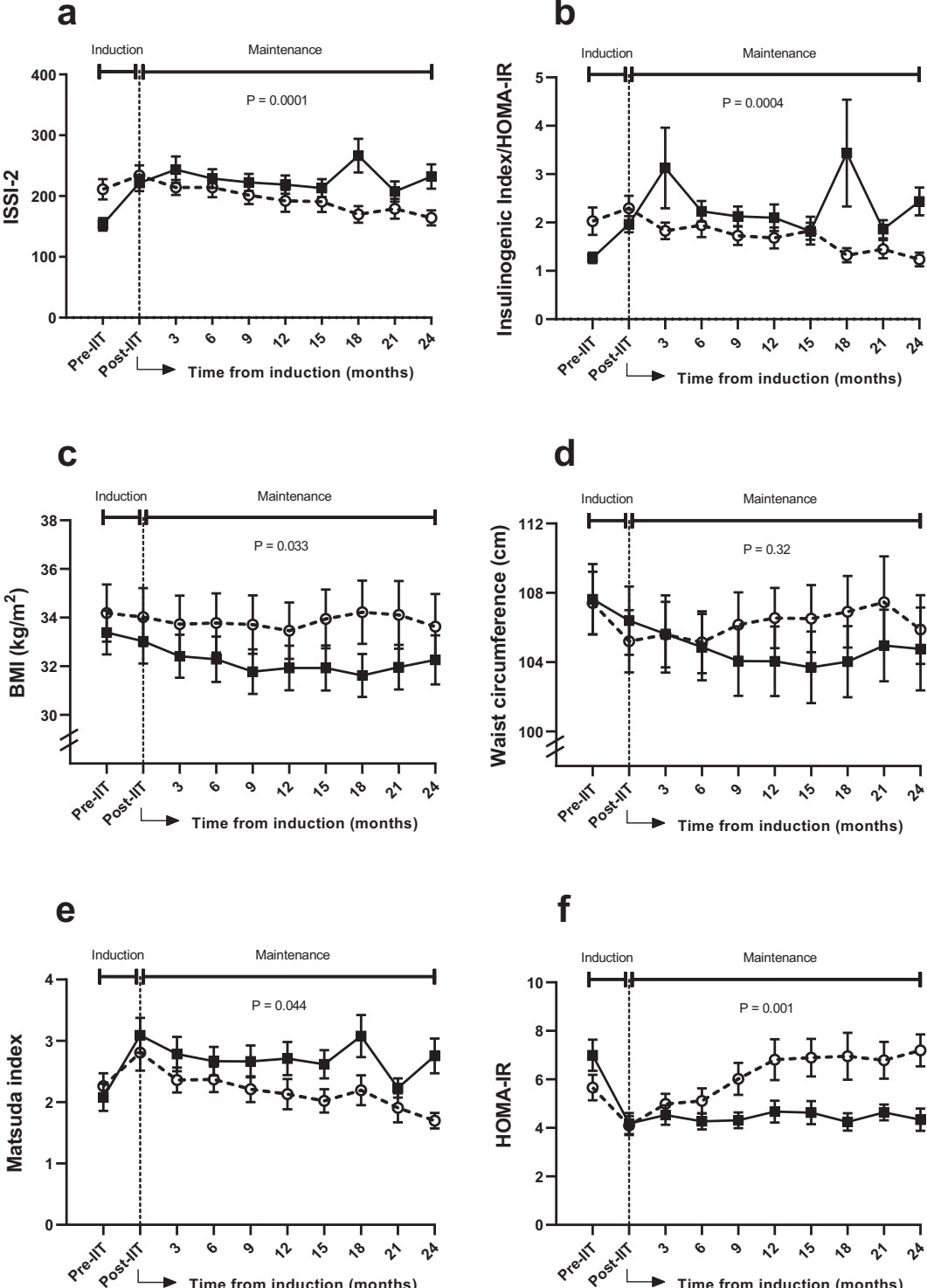

**Fig. 1 | Changes over time in metabolic variables.** (**a**) ISSI-2; (**b**) Insulinogenic index/HOMA-IR, (**c**) BMI, (**d**) Waist circumference, (**e**) Matsuda index; and (**f**) HOMA-IR, comparing those with stabilization of beta-cell function over 2-years and those without such stabilization. Post-IIT indicates the beginning of the maintenance phase that runs from 3-weeks to 24-months. *P*-values pertain to the interaction term between group and time during the maintenance phase from 3-weeks to 24-months. Data presented as mean ± standard error. Number of participants: no stabilization *n* = 44; stabilization *n* = 55.

**Table 4 | Logistic regression analyses of (dependent variable) stabilization of beta-cell function over 2 years: (A) core model derivation and (B) sensitivity analyses**

| Panel A: Core model derivation | Adjusted OR | 95%CI | P |
|---|---|---|---|
| **Model I: AUC 0.79; AIC 91.1** | | | |
| Age | 0.99 | (0.96, 1.03) | 0.75 |
| Duration of diabetes | 1.17 | (0.77, 1.78) | 0.46 |
| Log baseline ISSI-2 | 0.41 | (0.12, 1.42) | 0.16 |
| Change in ISSI-2 from baseline to 3-weeks | 1.01 | (1.00, 1.02) | **0.02** |
| Change in waist from 3-weeks to 2-years | 0.92 | (0.83, 1.02) | 0.10 |
| Change in Matsuda index from 3-weeks to 2-years | 1.80 | (1.01, 3.20) | **0.045** |
| **Model II: AUC 0.81; AIC 90.2** | | | |
| Age | 1.00 | (0.96, 1.04) | 0.95 |
| Duration of diabetes | 1.17 | (0.77, 1.77) | 0.47 |
| Log baseline ISSI-2 | 0.50 | (0.13, 1.85) | 0.30 |
| Change in ISSI-2 from baseline to 3-weeks | 1.01 | (1.00, 1.02) | **0.02** |
| Change in BMI from 3-weeks to 2-years | 0.65 | (0.42, 1.00) | **0.05** |
| Change in Matsuda index from 3-weeks to 2-years | 1.56 | (0.91, 2.68) | 0.11 |
| **Model III: AUC 0.88; AIC 83.7** | | | |
| Age | 1.00 | (0.96, 1.05) | 0.83 |
| Duration of diabetes | 1.03 | (0.66, 1.60) | 0.90 |
| Log baseline ISSI-2 | 0.20 | (0.04, 0.87) | **0.03** |
| Change in ISSI-2 from baseline to 3-weeks | 1.02 | (1.00, 1.03) | **0.005** |
| Change in BMI from 3-weeks to 2-years | 0.73 | (0.45, 1.19) | 0.21 |
| Change in HOMA-IR from 3-weeks to 2-years | 0.64 | (0.49, 0.83) | **0.0007** |
| **Panel B: Sensitivity analyses** | | | |
| **Model IV** | | | |
| Age | 1.01 | (0.96, 1.06) | 0.83 |
| Duration of diabetes | 1.86 | (0.81, 4.26) | 0.14 |
| Log baseline ISSI-2 | 0.18 | (0.02, 1.60) | 0.12 |
| Change in ISSI-2 from baseline to 3-weeks | 1.03 | (1.01, 1.05) | **0.004** |
| Change in BMI from 3-weeks to 2-years | 0.95 | (0.50, 1.82) | 0.88 |
| Change in HOMA-IR from 3-weeks to 2-years | 0.49 | (0.31, 0.78) | **0.003** |
| Average total physical activity over 2-years | 1.47 | (0.76, 2.87) | 0.25 |
| **Model V** | | | |
| Age | 0.98 | (0.93, 1.04) | 0.56 |
| Duration of diabetes | 0.95 | (0.56, 1.61) | 0.86 |
| Log baseline ISSI-2 | 0.17 | (0.02, 1.19) | 0.07 |
| Change in ISSI-2 from baseline to 3-weeks | 1.01 | (1.00, 1.03) | **0.02** |
| Change in BMI from 3-weeks to 2-years | 0.93 | (0.47, 1.85) | 0.83 |
| Change in HOMA-IR from 3-weeks to 2-years | 0.54 | (0.38, 0.77) | **0.0007** |
| Change in ALT from 3-weeks to 2-years | 0.92 | (0.86, 0.99) | **0.02** |

Two-sided *t*-test was performed for the estimated odds ratios with no adjustment for multiple comparisons. Bold indicates *p* ≤ 0.05.

3-weeks to 2-years supplanted the concomitant changes in Matsuda, BMI and waist in predicting sustained stabilization of beta-cell function. These data are suggestive of the specific importance of hepatic insulin sensitivity as a determinant of beta-cell function.

This emerging role of hepatic insulin sensitivity is supported by earlier observations. First, in women with recent gestational diabetes, worsening hepatic insulin resistance in the 1st year postpartum has been shown to independently predict declining beta-cell function, while concurrent changes in whole-body insulin sensitivity (Matsuda) and weight did not do so[12]. Second, in early T2DM, the change in HOMA-IR following 4-weeks of IIT has been associated with the concomitant change in beta-cell function[13]. Third, in the CORonary Diet Intervention with Olive oil and cardiovascular PREVention (CORDIO-PREV) trial, higher baseline hepatic insulin sensitivity emerged as a predictor of the likelihood of achieving remission of diabetes with Mediterranean and low-fat diets[14]. The current study extends these findings by demonstrating the potential importance of preserving hepatic insulin sensitivity for the durability of the beneficial beta-cell effects of initial short-term IIT.

It is notable that, while the change in HOMA-IR from 3-weeks to 2-years superseded the changes in Matsuda, BMI and waist, consideration of the concomitant change in ALT yielded the optimal model of sustained stabilization of beta-cell function (Model V in Table 4 Panel B). Moreover, the change in ALT from 3-weeks to 2-years emerged as a significant predictor in this model, alongside the change in HOMA-IR. These data further implicate the role of the liver in determining beta-cell health and suggest the presence of other relevant elements beyond hepatic insulin resistance as reflected by HOMA-IR. One possibility in this regard is hepatic fat, the impact of which warrants evaluation in future studies of the long-term trajectory of beta-cell function. Indeed, the emergence of the change in ALT as a significant predictor in Table 4 potentially may be pointing to the importance of regional fat deposition in determining this outcome. It is a limitation of this study is that participants were not assessed for concomitant metabolic dysfunction-associated fatty liver disease.

An additional limitation is the relatively modest sample size (*n* = 99), which precluded adjustment for multiple comparisons such that these analyses should be interpreted cautiously as hypothesis-generating. With a larger sample size, we speculate that the change in BMI during the maintenance phase may have emerged as an additional significant negative predictor of sustained stabilization of beta-cell function over two years in the logistic regression analyses in Table 4, along with its downstream implications for insulin sensitivity as reflected in the concomitant change in HOMA-IR. Another limitation is that insulin sensitivity/resistance and beta-cell function were measured with OGTT-based surrogate indices rather than clamp studies. However, it should be noted that these indices are all validated and established measures that have been widely applied in previous studies[3,11,15–21]. Moreover, the demands of clamp studies would have made it difficult for participants to undergo such assessments on 10 occasions over 2-years. In the current study, these serial assessments provided unique insight into the changes over time in metabolic function.

In conclusion, treatment with 3-weeks of induction IIT followed by metformin maintenance achieved stabilization of beta-cell function over 2-years in 55 of 99 participants. These individuals had poorer beta-cell function than their peers at baseline but experienced greater functional recovery thereof in response to induction IIT, suggestive of reversibility of beta-cell dysfunction. Their subsequent changes in HOMA-IR and ALT during the ensuing 2 years emerged as significant predictors of sustained stabilization of beta-cell function, coupled with the initial beta-cell response to induction IIT. Thus, the initial reversibility of beta-cell dysfunction during induction and the subsequent preservation of hepatic insulin sensitivity during maintenance are

physiologic changes associated with the sustained stabilization of beta-cell function in response to short-term IIT and metformin.

## Methods
This multi-centre clinical trial was approved by the research ethics boards of Mount Sinai Hospital (Toronto, ON), Western University (London, ON), and Hamilton Health Sciences (Hamilton, ON), which were sites where the study took place. The trial was registered at ClinicalTrials.Gov NCT02192424. It was conducted in accordance with Good Clinical Practice and the principles of the Declaration of Helsinki, and all participants provided written informed consent. The study protocol has been previously described in detail, along with reporting of the primary outcome[11].

### Study population
Inclusion criteria included age 30–80 years and duration of T2DM ≤ 5 years. Before the trial, participants could be treated with either metformin or lifestyle only. Exclusion criteria included treatment with anti-diabetic medication other than metformin, estimated glomerular filtration rate <50 ml/min, and known liver disease or transaminases > 2.5-fold above normal.

### Randomization and Interventions
Participants stopped metformin (if applicable) and fasted overnight prior to the baseline visit, at which they completed a 2 h 75 g oral glucose tolerance test (OGTT). At this visit, they received instruction on healthy lifestyle practices for managing T2DM[22,23] (which they were encouraged to continue throughout the trial) and were randomized (1:1) by computer-generated random allocation sequence to one of the following regimens: (i) 3-weeks of induction IIT followed by maintenance therapy with daily metformin thereafter for 2-years or (ii) the same regimen but with maintenance metformin supplemented with 2-week courses of IIT every 3-months (i.e. at 3-, 6-, 9-, 12-, 15-, 18- and 21-months).

These 2 treatment protocols have been described in detail previously in ref. 11. In brief, in both groups, induction IIT was administered as multiple daily injections consisting of bedtime insulin glargine and pre-prandial lispro. Doses were titrated to target fasting glucose between 4.0–6.0 mmol/l and 2 h postprandial glucose < 8 mmol/l on self-monitoring. On the last day of the 3-week induction, the final insulin dose was lispro prior to dinner (no bedtime glargine). Participants then fasted overnight and completed their 3-week OGTT the next morning.

Following the 3-week OGTT, participants began maintenance therapy consisting of either (i) metformin or (ii) metformin + intermittent IIT. In both arms, metformin was initiated at 1000 mg/day for 2 weeks, followed by 2000 mg/day or maximal tolerated dose thereafter. In the intermittent IIT arm, a 2-week course of IIT was administered every 3-months with the same glucose targets as during induction, as previously described in detail in ref. 11. These 2-week courses of IIT were started at study visits at 3-, 6-, 9-, 12-, 15-, 18- and 21-months, following completion of an OGTT at each of these visits. A final OGTT was done at 24-months.

### Physiologic Indices on OGTT
The serial 2 h 75 g OGTTs enabled assessment of physiologic indices reflecting insulin sensitivity/resistance and beta-cell function. Participants fasted overnight prior to each OGTT, with metformin held on the morning of the test. During each OGTT, venous blood samples were drawn at fasting and at 30-, 60-, 90- and 120-min post-challenge. Glucose, insulin and C-peptide were measured from these samples, as previously described in ref. 11. Whole-body insulin sensitivity was assessed by Matsuda index[15] and hepatic insulin resistance was assessed by Homeostasis Model Assessment (HOMA-IR)[16]. Beta-cell function was assessed in 4 ways, with the primary measure being the Insulin Secretion-Sensitivity Index-2 (ISSI-2) (the baseline-adjusted value of which at 2-years was the primary outcome of the trial). ISSI-2 is a validated OGTT-based measure of beta-cell compensation that is analogous to the disposition index from the intravenous glucose tolerance test, against which it has been directly validated[17,18]. The other 3 measures of beta-cell function were (i) insulinogenic index/HOMA-IR, (ii) $\Delta Cpep_{0-120}/\Delta gluc_{0-120} \times$ Matsuda index and (iii) $\Delta ISR_{0-120}/\Delta gluc_{0-120} \times$ Matsuda index (where ISR is the pre-hepatic insulin secretion rate determined by C-peptide deconvolution), with these indices calculated as previously described in refs. 3,11.

### Outcomes
The primary and secondary outcomes of the trial have been previously reported in detail[11]. The current secondary analysis focused on sustained stabilization of beta-cell function, which was defined by having higher ISSI-2 at 2-years (or last study visit) than at baseline. Of the 108 study participants, there were 9 individuals whose last study visit occurred at <12-months, representing a duration of follow-up that was considered insufficient for determination of sustained stabilization of beta-cell function. The current analysis was thus performed in the 99 participants in whom sustained stabilization of beta-cell function could be assessed.

### Statistical Analyses
Statistical analyses were conducted with R 4.2.2, with two-tailed $P$-values < 0.05 considered statistically significant. Continuous variables were tested for normality of distribution, with natural log transformation of skewed variables conducted where necessary. The study population was first stratified into those who did and did not achieve stabilization of beta-cell function. Characteristics of the stabilization and non-stabilization groups at baseline, after 3-weeks of induction IIT and after 2-years of maintenance therapy were compared by Analysis of Variance (normally distributed variables) or Kruskal-Wallis test (skewed variables), or Chi-Square test (categorical variables)(Tables 1–3). The daily profile of mean capillary blood glucose during induction IIT was determined for each group (Supplementary Fig. 1). The time to initial loss of stabilization of beta-cell function (defined by 2 consecutive visits with ISSI-2 lower than at baseline) was compared between treatment arms by log-rank test (Supplementary Fig. 2). The glucose response on the OGTT was compared between the stabilization and non-stabilization groups at 6 study visits, with differences at each time-point on the test assessed by $t$-test (Supplementary Fig. 3). The longitudinal changes over time in ISSI-2, insulinogenic index/HOMA-IR, BMI, waist circumference, Matsuda index, and HOMA-IR were compared between the groups by generalized least squares (GLS) regression, wherein the interaction effect between group and time was examined (Fig. 1). Logistic regression analyses were performed to develop a core model of predictors of stabilization of beta-cell function over 2-years amongst age, duration of diabetes, log baseline ISSI-2, change in ISSI-2 during induction, and changes in waist/BMI and Matsuda/HOMA-IR during maintenance (Table 4 Panel A). Area-under-the-curve (AUC) and Akaike Information Criterion (AIC) enabled comparison of non-nested models I, II and III with respect to prediction accuracy and goodness-of-fit, with highest AUC and lowest AIC identifying the core model. Sensitivity analyses were performed by adding total physical activity over 2 years (assessed by Baecke questionnaire[24,25]) or change in ALT during maintenance to the core model, with likelihood ratio tests performed to compare nested regression models (Table 4 Panel B).

### Reporting summary
Further information on research design is available in the Nature Portfolio Reporting Summary linked to this article.

## Data availability
De-identified data can be made available under restricted access from the corresponding author (Ravi.Retnakaran@SinaiHealth.ca), for

academic purposes, subject to a material transfer agreement and approval of the Mount Sinai Hospital Research Ethics Board. Individual participant data that underlie the results reported in this article can be made available by this mechanism, after de-identification, to achieve the aims in the approved proposal. Access is controlled in this way because of the clinical nature of the data. The study protocol can also be made available in this way. This data access mechanism will be available beginning 9 months and ending 36 months following publication of this article. We will attempt to respond to requests within 3 months, pending Research Ethics Board capacity to do so within this time frame. Source data for figures have been provided with this paper. Source data are provided with this paper.

## Code availability
The code used for the statistical analyses is available at Github at the following link: https://github.com/rretnakaran/Determinants-of-Beta-cell-Function.git

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

## Acknowledgements
RR holds the Boehringer Ingelheim Chair in Beta-cell Preservation, Function and Regeneration at Mount Sinai Hospital. SBH holds the Diabetes Canada Chair in Diabetes Management at Western University. SMR holds the Brian W. Gilbert Chair in Primary Health Care at Western University. HCG holds the McMaster-Sanofi Population Health Institute Chair in Diabetes Research and Care. This study was supported by the Canadian Institutes of Health Research (CIHR) (MOP 133701) (RR). The funder had no role in study design, data collection and analysis, or writing of the manuscript.

## Author contributions
R.R. and B.Z. designed the study. R.R., A.E., S.B.H., S.R., H.C.G., N.M., C.K.K. and B.Z. implemented the study and acquired the data. R.R. led overall study implementation. A.E. oversaw research coordination. J.P. performed the statistical analyses. R.R. wrote the manuscript. R.R. and J.P. verified the data. All authors contributed to interpretation of the data and critical revision of the manuscript. All authors approved the manuscript. R.R. is the guarantor of this work.

## Competing interests
R.R. reports grants from Boehringer Ingelheim, grants and personal fees from Novo Nordisk, personal fees from Sanofi, personal fees from Eli Lilly, outside the submitted work. SBH reports grants and personal fees from Sanofi, grants and personal fees from Novo Nordisk, grants and

personal fees from AstraZeneca, personal fees from Eli Lilly, personal fees from Janssen, grants and personal fees from Abbott, outside the submitted work. S.M.R. reports personal fees and non-financial support from Boehringer Ingelheim, Janssen, Eli Lilly, Sanofi, Bayer and Astra-Zeneca; grants, personal fees and non-financial support from Abbott, and Novo Nordisk; personal fees from Western University, Schulich School of Medicine and Dentistry and The Federation of Medical Women of Canada; grants from the Canadian Institute of Health Research, outside the submitted work. N.M. reports grants and non-financial support from AstraZeneca, grants and non-financial support from Merck, grants and non-financial support from Sanofi, outside the submitted work. H.C.G. reports research grants from Eli Lilly, AstraZeneca, Merck, Novo Nordisk and Sanofi; honoraria for speaking from Boehringer Ingelheim, Eli Lilly, Novo Nordisk, Sanofi, DKSH, and Zuellig; and consulting fees from Abbott, Covance, Eli Lilly, Novo Nordisk, Sanofi, Pfizer and Kowa. C.K.K. reports research grants from Boehringer Ingelheim. J.P., A.E., and B.Z. have nothing to disclose.
