## [Peer Review File · Nature Communications]

Determinants of Sustained Stabilization of Beta-cell Function Following Short-term Insulin Therapy in Type 2 DiabetesREVIEWER COMMENTS

Reviewer #1 (Remarks to the Author):

Thank you for inviting me to review the manuscript. This is a post hoc analysis of the trial data from the RESET-IT Main trial. It was aimed to elucidate factors which dominated the sustained stabilization of pancreatic beta cell function in response to short-term intensive insulin therapy (IIT) followed by metformin with/without intermittent IIT in early type 2 diabetes. The authors concluded that the initial reversibility of beta cell function during induction and subsequent preservation of hepatic insulin sensitivity during maintenance were associated with sustained stabilization of beta cell function.

The research question is of clinical value and should provide insight into the mechanism of remission/preservation of beta cell function in early type 2 diabetes. However, I have some comments on the result interpretation, as follows:

1. Change of BMI. Loss of weight has been an established factor of better metabolic control and relief of insulin resistance. In Table 2, it was clear that patients with sustained stabilization of beta cell function at 2-yrs had a significantly greater BMI reduction. Moreover, for logistic regression Model II, though it was not the best available model, changes in BMI also appeared to be an independent factor predictive of the preservation of beta cell function. And thus, I think the conclusion of this study should be taken with more caution. Please elaborate discussion regarding BMI and the potential mechanism behind the findings in the study.
2. It is reasonable that sustained stabilization of beta cell function is associated with relieved hepatic insulin resistance. Metabolic dysfunction-associated fatty liver disease (MAFLD) is well-known to be associated with hepatic insulin resistance and type 2 diabetes. So, were the participants investigated for their concomitance with MAFLD? If yes, did the concomitance of MAFLD related to the stabilization of beta cell function or the change of ALT, which was included in Model V?
3. Was there any ethnic difference regarding the sustained stabilization of beta cell

function?

Minor points:

In Table 3, Panel B, there are two "Model VI".

Reviewer #2 (Remarks to the Author):

This study is a report on the mechanistical findings produced by the RESET-IT trial that has been published earlier. Focus of this trial was the investigation of stabilization of pancreatic beta cell function in response to early IIT in early type 2 diabetes. RESET-IT demonstrated stabilized beta-cell function for 2-years in some patients but not others, but the addition of intermittent courses of IIT did not further enhance the beneficial effect on beta-cell function achieved with initial induction IIT followed by metformin maintenance.

In this report the authors sought to identify and characterize patient-individual determinants that may be associated with stabilization of pancreatic beta cell function by analysis of data collected within the RESET-IT trial.

The authors used state of the art parameters for determination of beta cell function and insulin sensitivity such as Insulin Secretion-Sensitivity Index-2 (ISSI-2), insulinogenic index/HOMA-IR, $\Delta\text{Cpeptide}_{0-120}/\Delta\text{glucose}_{0-120}\times\text{Matsuda}$, and $\Delta\text{ISR}_{0-120}/\Delta\text{gluc}_{0-120}\times\text{Matsuda}$.

The authors found that those patients (a little more than half of the trial population) who achieved sustained preservation of pancreatic beta cell function by the intervention had poorer beta-cell function at baseline, but had greater beta-cell functional recovery. Further determinants in these patients for sustained response were lesser rise in BMI, HOMA-IR and ALT during maintenance period, and change in ISSI-2 during induction IIT, changes in HOMA-IR and ALT during maintenance therapy from 3-weeks to 2-years.

This analysis adds important clinically relevant information on the effect of early intensive insulin treatment in patients with type 2 diabetes on the propensity of preservation of endogenous insulin biosynthesis. These results go beyond the knowledge already present on this type of intervention, and adds information that may be helpful in translation into clinical practice.

Reviewer #3 (Remarks to the Author):

I read this paper with great interest. The authors conducted an analysis based on their previous RESETIT Main to elucidate determinants of sustained stabilization of beta-cell function in response to short-term IIT and metformin in early T2DM. They found that initial reversibility of beta-cell dysfunction during induction and subsequent preservation of hepatic insulin sensitivity during maintenance are associated with the sustained stabilization of beta-cell function in response to short-term IIT and metformin.

This is a very well-written paper, with significant scientific importance, and is in the journal scope. However, I had some concerns and comments, mostly related to the analytical approach and report (comments with * indicates major issues):

1. Page 2, Abstract: Line 14. Please give "ALT" a full term before using its abbreviation

2*. Page 7, Results: Line 13-21. I am a little confused. Are these p-values for interaction between group and time, or main effect of group? For example, Figure 1-D, it seems like there is an interaction between group and time, but I am surprised that the p-value is 0.65.

3*. Figure 1: Is the "Post-IIT" the same as the beginning of "3-weeks"? If so, please specify clearly. If not, please also provide a marker for "3-weeks".

4*. Page 8, Results: Line 1-12. Sorry, I did not fully follow up the logic of model fitting and variable selection steps across Model I to Model III. Are you primarily focusing on model improvement to identify independent determinants of the stabilization of beta-cell function at 2-years? While statistical justification is important, both clinical relevance and logical modification should not be avoided. For example, why in Model III, we retain "Change in BMI from 3-weeks to 2-years" instead of "Change in waist from 3-weeks to 2-years"? I knew that you previously substituted it by "BMI" since "waist" was not statistically significant (Model I to Model II), but in both Model I and II, "Change in Matsuda index from 3-weeks to 2-years" was included in the models. However, in Model III, "Change in HOMA-IR from 3-weeks to 2-years" was included in the model, thus, you would not know if "Change in waist from 3-weeks to 2-years" will provide a better model improvement or not.

5*. Page 16, Statistical Analyses: Line 8-10. I have several questions related to the analysis and definitions here.

a) Is it possible that the subject may loss of stabilization of beta-cell function multiple times during the 2-year period? If so, should Online Fig 2 be better referred to "time to first loss of

stabilization of beta-cell function”?

b) Is it possible that the subject may lose stabilization of beta-cell function but resume back to stabilization of beta-cell function again during the 2-year period?

c) Related to the above: since sustained stabilization of beta-cell function was defined by having higher ISSI-2 at 2-years (or last study visit) than at baseline, is it possible that the subject may be classified as sustained stabilization of beta-cell function but may still lose stabilization of beta-cell function during the 2-year period?

6. Page 30, Table 3: Will you please also provide AUC and AIC for Panel B: Sensitivity analyses models?

7. In both Table 1 and Table 2, you performed so many group comparisons among 30+ variables. Will you consider p-value adjustment for multiple comparisons? I think this is particularly important for Table 2 (these are your endpoint outcome variables). Or at least, you need to address it as a limitation for cautious interpretation (given only a total of 99 subjects among 30+ comparisons).

We thank the reviewers for their kind comments.

REVIEWER 1 COMMENTS

GENERAL COMMENT: *“Thank you for inviting me to review the manuscript. This is a post hoc analysis of the trial data from the RESET-IT Main trial. It was aimed to elucidate factors which dominated the sustained stabilization of pancreatic beta cell function in response to short-term intensive insulin therapy (IIT) followed by metformin with/without intermittent IIT in early type 2 diabetes. The authors concluded that the initial reversibility of beta cell function during induction and subsequent preservation of hepatic insulin sensitivity during maintenance were associated with sustained stabilization of beta cell function.*

The research question is of clinical value and should provide insight into the mechanism of remission/preservation of beta cell function in early type 2 diabetes. However, I have some comments on the result interpretation, as follows.”

RESPONSE: We thank the reviewer for the kind comments and have addressed each of the specific comments in turn below.

I. COMMENT: *“Change of BMI. Loss of weight has been an established factor of better metabolic control and relief of insulin resistance. In Table 2, it was clear that patients with sustained stabilization of beta cell function at 2-yrs had a significantly greater BMI reduction. Moreover, for logistic regression Model II, though it was not the best available model, changes in BMI also appeared to be an independent factor predictive of the preservation of beta cell function. And thus, I think the conclusion of this study should be taken with more caution. Please elaborate discussion regarding BMI and the potential mechanism behind the findings in the study.”*

RESPONSE: We thank the reviewer for this comment and fully concur that weight loss is an established contributor to the relief of insulin resistance. In the logistic regression analyses in Table 3, change in BMI during the maintenance phase from 3-weeks to 2-years achieved statistical significance (at $p=0.05$) as a negative predictor of sustained stabilization of beta-cell function in Model II (alongside change in Matsuda index) but not in Model III (where change in Matsuda index was replaced with the change in HOMA-IR). We believe that change in BMI likely did not reach significance in the latter model because of the modest sample size ($n=99$), wherein it was supplanted by potentially the most direct downstream factor (hepatic insulin resistance) linking change in BMI to the outcome (i.e. whereas whole-body insulin sensitivity as measured by change in Matsuda index allowed it to reach significance in Model II). We suspect that, with a larger sample size, change in BMI may have indeed achieved significance as a negative predictor of sustained stabilization of beta-cell function. We now note this point in the Discussion with text as follows: *“An additional limitation is the relatively modest sample size ($n=99$), which precluded adjustment for multiple comparisons such that these analyses should be interpreted cautiously as hypothesis-generating. With a larger sample size, we speculate that the change in BMI during the maintenance phase may have emerged as an additional significant negative predictor of sustained stabilization of beta-cell function over two years in the logistic regression analyses in Table 3, along with its downstream implications for insulin sensitivity as reflected in the concomitant change in HOMA-IR.”* (Page 12 Paragraph 3 Lines 1-7)

The presumed mechanism is also now addressed in the Discussion with the following text:

“Conversely, weight gain may increase insulin resistance and hasten beta-cell demise. In this context, it is notable that, during maintenance from 3-weeks to 2-years, there were differential changes over time in BMI between the stabilization and non-stabilization groups, with evidence of a decrease in the stabilization group (Figure 1 Panel C). However, far more striking were the differential changes in insulin sensitivity/resistance, possibly reflecting downstream sequelae of the changes in BMI.” (Page 11 Paragraph 1 Lines 3-9)

2. **COMMENT:** *“It is reasonable that sustained stabilization of beta cell function is associated with relieved hepatic insulin resistance. Metabolic dysfunction-associated fatty liver disease (MAFLD) is well-known to be associated with hepatic insulin resistance and type 2 diabetes. So, were the participants investigated for their concomitance with MAFLD? If yes, did the concomitance of MAFLD related to the stabilization of beta cell function or the change of ALT, which was included in Model V?”*

RESPONSE: We are grateful to the reviewer for this point and fully concur that evaluation of participants for concomitant metabolic dysfunction-associated fatty liver disease would be of interest. Unfortunately, however, it is a limitation of this study that such evaluation was not performed. We now note this limitation in the Discussion, with the following text: *“One possibility in this regard is hepatic fat, the impact of which warrants evaluation in future studies of the long-term trajectory of beta-cell function. Indeed, the emergence of the change in ALT as a significant predictor in Table 3 potentially may be pointing to the importance of regional fat deposition in determining this outcome. It is a limitation of this study is that participants were not assessed for concomitant metabolic dysfunction-associated fatty liver disease.”* (Page 12 Paragraph 2 Lines 7-12)

3. **COMMENT:** *“Was there any ethnic difference regarding the sustained stabilization of beta cell function?”*

RESPONSE: We are grateful to the reviewer for this comment. There were no significance differences in ethnicity between those with and without stabilization of beta-cell function, as shown in Table 1 wherein ethnicity was classified as white, South Asian, East Asian or other (p=0.89). Similarly, there were no significant differences between the stabilization and non-stabilization groups when ethnicity was classified as either white or non-white (p=0.99) (data not shown).

4. **COMMENT:** *“Minor points: In Table 3, Panel B, there are two “Model VI”.”*

RESPONSE: We thank the reviewer for catching this error, which we have now corrected in Table 3 Panel B.

REVIEWER 2 COMMENTS

GENERAL COMMENT: *“This study is a report on the mechanistical findings produced by the RESET-IT trial that has been published earlier. Focus of this trial was the investigation of stabilization of pancreatic beta cell function in response to early IIT in early type 2 diabetes. RESET-IT demonstrated stabilized beta-cell function for 2-years in some patients but not others, but the addition of intermittent courses of IIT did not further enhance the beneficial effect on beta-cell function achieved with initial induction IIT followed by metformin maintenance.*

In this report the authors sought to identify and characterize patient-individual determinants that may be associated with stabilization of pancreatic beta cell function by analysis of data collected within the RESET-IT trial.

The authors used state of the art parameters for determination of beta cell function and insulin sensitivity such as Insulin Secretion-Sensitivity Index-2 (ISSI-2), insulinogenic index/HOMA-IR, Δ Cpeptide₀₋₁₂₀/ Δ glucose₀₋₁₂₀×Matsuda, and Δ ISR₀₋₁₂₀/ Δ gluc₀₋₁₂₀×Matsuda.

The authors found that those patients (a little more than half of the trial population) who achieved sustained preservation of pancreatic beta cell function by the intervention had poorer beta-cell function at baseline, but had greater beta-cell functional recovery. Further determinants in these patients for sustained response were lesser rise in BMI, HOMA-IR and ALT during maintenance period, and change in ISSI-2 during induction IIT, changes in HOMA-IR and ALT during maintenance therapy from 3-weeks to 2-years.

This analysis adds important clinically relevant information on the effect of early intensive insulin treatment in patients with type 2 diabetes on the propensity of preservation of endogenous insulin biosynthesis. These results go beyond the knowledge already present on this type of intervention, and adds information that may be helpful in translation into clinical practice.”

RESPONSE: We thank the reviewer for the kind comments.

REVIEWER 3 COMMENTS

GENERAL COMMENT: *“I read this paper with great interest. The authors conducted an analysis based on their previous RESET IT Main to elucidate determinants of sustained stabilization of beta-cell function in response to short-term IIT and metformin in early T2DM. They found that initial reversibility of beta-cell dysfunction during induction and subsequent preservation of hepatic insulin sensitivity during maintenance are associated with the sustained stabilization of beta-cell function in response to short-term IIT and metformin.*

*This is a very well-written paper, with significant scientific importance, and is in the journal scope. However, I had some concerns and comments, mostly related to the analytical approach and report (comments with * indicates major issues):”*

RESPONSE: We thank the reviewer for the kind comments and have addressed each of the specific comments below in turn.

1. COMMENT: *“Page 2. Abstract: Line 14. Please give “ALT” a full term before using its abbreviation.”*

RESPONSE: We thank the reviewer for this comment and have now identified ALT as alanine aminotransferase in the Abstract on Page 2 Line 14.

2. COMMENT: *“* Page 7, Results: Line 13-21. I am a little confused. Are these p-values for interaction between group and time, or main effect of group? For example, Figure 1-D, it seems like there is an interaction between group and time, but I am surprised that the p-value is 0.65.”*

RESPONSE: We thank the reviewer for this comment. The p-values in Figure 1 had been those pertaining to the group effect after adjustment for time. However, in considering the reviewer’s comment, we recognized that presentation of the p-value for the interaction between group and time would provide better insight into differential changes over time between the groups. We have thus revised Figure 1 to now show the p-value for the interaction between group and time (i.e. the plots in Figure 1 are unchanged; the only difference is that we now present the p-value

for group-time interaction). Accordingly, we have now revised the caption to Figure 1 (Page 30), with the following text: “*P-values pertain to the interaction between group and time during the maintenance phase from 3-weeks to 24-months.*” We have also revised the description in the Results section with the following text: “*To determine whether these measures were changing differentially over time between the 2 groups, we next performed generalized least squares regression analyses (Figure 1). As anticipated, there were differential changes over time in ISSI-2 ($p<0.001$) and insulinogenic index/HOMA-IR ($p<0.001$) during maintenance from 3-weeks to 2-years, with the non-stabilization group showing loss of beta-cell function (Figure 1 Panels A and B). Of note, BMI ($p=0.033$) also changed differentially between the groups, with decrease over time evident in the stabilization group, although waist circumference ($p=0.32$) did not show such differential change (Figure 1 Panels C and D). However, there were differential changes in both whole-body sensitivity (Matsuda index: $p=0.044$) and hepatic insulin resistance (HOMA-IR: $p=0.001$), with clear deterioration of each in the non-stabilization group (Figure 1 Panels E and F).*” (Page 7 Paragraph 3 Lines 5-15)

3. **COMMENT:** “* Figure 1: Is the “Post-IIT” the same as the beginning of “3-weeks”? If so, please specify clearly. If not, please also provide a marker for “3-weeks”.”

RESPONSE: We are grateful to the reviewer for this comment and now clarify the following in the caption to Figure 1: “*Post-IIT indicates the beginning of the maintenance phase that runs from 3-weeks to 24-months.*” (Page 30)

4. **COMMENT:** “* Page 8, Results: Line 1-12. Sorry, I did not fully follow up the logic of model fitting and variable selection steps across Model I to Model III. Are you primarily focusing on model improvement to identify independent determinants of the stabilization of beta-cell function at 2-years? While statistical justification is important, both clinical relevance and logical modification should not be avoided. For example, why in Model III, we retain “Change in BMI from 3-weeks to 2-years” instead of “Change in waist from 3-weeks to 2-years”? I knew that you previously substituted it by “BMI” since “waist” was not statistically significant (Model I to Model II), but in both Model I and II, “Change in Matsuda index from 3-weeks to 2-years” was included in the models. However, in Model III, “Change in HOMA-IR from 3-weeks to 2-years” was included in the model, thus, you would not know if “Change in waist from 3-weeks to 2-years” will provide a better model improvement or not.”

RESPONSE: We thank the reviewer for this comment and fully concur that both statistical justification and clinical relevance are factors to consider in model construction. As the reviewer notes, the 3 core models shown in Table 3 Panel A evaluate change in BMI with change in Matsuda (Model II) and change in HOMA-IR (Model III), respectively, while considering change in waist with only change in Matsuda (Model I).

We did, however, also do a model that tested change in waist with change in HOMA-IR (i.e. substituting change in HOMA-IR for change in Matsuda in Model I), which revealed that change in waist from 3-weeks to 2-years was not a significant predictor of stabilization of beta-cell function over 2-years (adjusted OR=0.92 [95% CI 0.83-1.02], $p=0.11$). The only significant predictors in that model were the same as those in Model III: log baseline ISSI-2 (aOR=0.20 [0.04-0.87], $p=0.03$), change ISSI=2 from baseline to 3-weeks (aOR=1.02 [1.00–1.03], $p=0.005$), and change in HOMA-IR from 3-weeks to 2-years (aOR=0.60, [0.46-0.78], $p<0.001$). We now report this model in the Results section with the following text: “*The significant predictors in Model III were unchanged when the change in BMI from 3-weeks to 2-years was*

replaced with the concomitant change in waist circumference (data not shown).” (Page 8 Paragraph 1 Lines 14-16).

The sequence of models presented in Table 3 Panel A was driven by both clinical and statistical considerations. From a clinical perspective, we thought that it would be important to show both models for change in BMI because, unlike waist, BMI showed evidence of differential change between the groups during the maintenance phase (e.g. Figure 1 Panel C). For that reason, when change in waist was not significant in Model I, it was replaced by change in BMI in Model II (while keeping change in Matsuda in both models). When change in Matsuda was then not significant in Model II, it was replaced by change in HOMA-IR in Model III (while keeping change in BMI in the model).

5. **COMMENT:** “* Page 16, Statistical Analyses: Line 8-10. I have several questions related to the analysis and definitions here.

a) Is it possible that the subject may loss of stabilization of beta-cell function multiple times during the 2-year period? If so, should Online Fig 2 be better referred to “time to first loss of stabilization of beta-cell function”?

b) Is it possible that the subject may lose stabilization of beta-cell function but resume back to stabilization of beta-cell function again during the 2-year period?

c) Related to the above: since sustained stabilization of beta-cell function was defined by having higher ISSI-2 at 2-years (or last study visit) than at baseline, is it possible that the subject may be classified as sustained stabilization of beta-cell function but may still lose stabilization of beta-cell function during the 2-year period?”

RESPONSE: We thank the reviewer for these comments, which will be address in turn:

(a) The reviewer is quite correct that Online Figure 2 is showing the time to initial loss of stabilization of beta-cell function, as defined by 2 consecutive visits with ISSI-2 lower than at baseline in the two treatment groups in the trial. We have now clarified this point in the Methods section, with text as follows: “*The time to initial loss of stabilization of beta-cell function (defined by 2 consecutive visits with ISSI-2 lower than at baseline) was compared between treatment arms by log-rank test (Online Fig. 2).*” (Page 16 Paragraph 3 Lines 9-11). We have also changed the caption to Online Figure 2 to the following: “*Online Fig 2: Kaplan-Meier curves showing time to initial loss of stabilization of beta-cell function between the metformin and metformin + IIT arms (defined by 2 consecutive visits with ISSI-2 lower than at baseline).*” The point of Online Figure 2 was to show that treatment group in the trial did not affect time to initial loss of stabilization of beta-cell function (having already shown in Table 1 that treatment group did not differ significantly between those who had sustained stabilization of beta-cell function at 2-years and those who did not).

(b) The main outcome of interest in this manuscript is sustained stabilization of beta-cell function at 2-years (or last study visit), which was defined by having higher ISSI-2 at 2-years (or last study visit) than at baseline (as indicated on lines 1-3 of the Outcomes subsection of the Methods on Page 16). Thus, this outcome is distinct from initial loss of stabilization such that both could occur in an individual.

(c) Indeed, based on the two definitions above, it is possible to have had initial loss of stabilization (as shown in Online Figure 2) but still achieve sustained stabilization of beta-cell function at 2-years or the last study visit. Of the 55 individuals classified as

having sustained stabilization of beta-cell function, there were 9 who did so after first meeting the definition of initial loss of stabilization.

6. **COMMENT:** “Page 30, Table 3: Will you please also provide AUC and AIC for Panel B: Sensitivity analyses models?”

RESPONSE: We are grateful to the reviewer for this comment. The likelihood ratio test is used to compare the sensitivity analyses (Models IV and V) with Model III because of their relationship as nested models (i.e. the predictors in Model III represent a subset of the predictors in Model IV and Model V). In the setting of nested models, the likelihood ratio test is a formal hypothesis test for the added variable and provides more stringent results with significance. In contrast, AIC and AUC serve as informal measures for comparing non-nested models (as in the comparisons amongst Models I, II and III). We have now clarified the application of the AIC/AUC and likelihood ratio tests in Table 3 in the Methods section, with text as follows: “Area-under-the-curve (AUC) and Akaike Information Criterion (AIC) enabled comparison of non-nested models I, II and III with respect to prediction accuracy and goodness-of-fit, with highest AUC and lowest AIC identifying the core model. Sensitivity analyses were performed by adding total physical activity over 2 years (assessed by Baecke questionnaire^{24,25}) or change in ALT during maintenance to the core model, with likelihood ratio tests performed to compare nested regression models (Table 3 Panel B).” (Page 17 Paragraph 1 Lines 9-15)

7. **COMMENT:** “In both Table 1 and Table 2, you performed so many group comparisons among 30+ variables. Will you consider p-value adjustment for multiple comparisons? I think this is particularly important for Table 2 (these are your endpoint outcome variables). Or at least, you need to address it as a limitation for cautious interpretation (given only a total of 99 subjects among 30+ comparisons).”

RESPONSE: We thank the reviewer for this comment and concur that the lack of adjustment for multiple comparisons is a limitation of this study in the context of the relatively modest sample size (n=99). We now note this limitation and the resultant need for cautious interpretation of these data in the Discussion, with text as follows: “An additional limitation is the relatively modest sample size (n=99), which precluded adjustment for multiple comparisons such that these analyses should be interpreted cautiously as hypothesis-generating. With a larger sample size, we speculate that the change in BMI during the maintenance phase may have emerged as an additional significant negative predictor of sustained stabilization of beta-cell function over two years in the logistic regression analyses in Table 3, along with its downstream implications for insulin sensitivity as reflected in the concomitant change in HOMA-IR.” (Page 12 Paragraph 3 Lines 1-7)

REVIEWERS' COMMENTS

Reviewer #1 (Remarks to the Author):

I am satisfied with the responses from the authors. And I have no further comments.

Reviewer #2 (Remarks to the Author):

The authors have fully addressed the recommendations of this reviewer.

Reviewer #3 (Remarks to the Author):

Overall, I am satisfied with the revised manuscript.

Two minor things:

1. LINE 102: you missed a right parenthesis
2. Figure 1: Please footnote what is the error bar stands for (standard error, or standard deviation?)

Congratulation!

REVIEWER 1:

GENERAL COMMENT: *“I am satisfied with the responses from the authors. And I have no further comments.”*

RESPONSE: We thank the reviewer for the kind comment.

REVIEWER 2:

GENERAL COMMENT: *“The authors have fully addressed the recommendations of this reviewer.”*

RESPONSE: We thank the reviewer for the kind comment.

REVIEWER 3:

GENERAL COMMENT: *“Overall, I am satisfied with the revised manuscript. Two minor things:”*

RESPONSE: We thank the reviewer for the kind comment and have addressed each of the specific comments below in turn.

1. COMMENT: *“LINE 102: you missed a right parenthesis.”*

RESPONSE: We thank the reviewer for catching this omission and have added the parenthesis as suggested (Page 4 Paragraph 2 Line 6)

2. COMMENT: *“Figure 1: Please footnote what is the error bar stands for (standard error, or standard deviation?)”*

RESPONSE: We thank the reviewer for this comment and, as suggested, have now indicated in the legend to Figure 1 that the error bar indicates standard error.